# An Assessment of the Effectiveness and Safety of ULTRACOL100 as a Device for Restoring Skin in the Nasolabial Fold Region

**Thuy-Tien Thi Trinh** [1,2,†], **Pham Ngoc Chien** [1,2,†], **Linh Thi Thuy Le** [1,3], **Nguyen Ngan-Giang** [1,4], **Pham Thi Nga** [1,2], **Sun-Young Nam** [1,*] **and Chan-Yeong Heo** [1,2,3,*]

1   Department of Plastic and Reconstructive Surgery, Seoul National University Bundang Hospital, Seongnam 13620, Republic of Korea; thuytienbiotech@gmail.com (T.-T.T.T.)
2   Korea Skin Clinical Research Center, Seongnam 13620, Republic of Korea
3   Department of Biomedical Science, College of Medicine, Seoul National University, Seoul 03080, Republic of Korea
4   Department of Medical Device Development, College of Medicine, Seoul National University, Seoul 03080, Republic of Korea
*   Correspondence: 99261@snubh.org (S.-Y.N.); lionheo@snu.ac.kr (C.-Y.H.)
†   These authors contributed equally to this work.

**Abstract:** One of the most notable signs of an aging face is the nasolabial folds (NLFs), which often diminish emotional well-being and self-confidence. To address this concern, many people seek solutions to improve their appearance, often turning to fillers. The ULTRACOL100 device, a tissue restoration material, has been previously investigated and shown to exhibit significant efficacy in both in vitro and in vivo studies. In this research, we aim to explore the safety and effectiveness of the clinical trial of ULTRACOL100 in improving the skin in the NLF area over an 8-week observation period. Male and Female adults with nasolabial folds received two injections of ULTRACOL100, with a 4-week interval between treatments, on one side of their faces. On the other side, they received control materials (REJURAN®, JUVELOOK®, or HYRONT®). The assessment of skin improvement in the nasolabial fold area for each subject took place before and four weeks after each application. Various skin parameters, such as roughness, elasticity, moisture, transparency, trans-epidermal water loss, tone, radiance, skin pore size, and skin density, were measured to evaluate the outcomes. The application of the ULTRACOL100 device significantly reduced the skin roughness, the trans-epidermal water loss, and the skin pore size and increased the skin's elasticity and internal elasticity, as well as the skin's moisture, transparency, skin tone, radiance, and density. This study comprehensively investigates the effectiveness and safety of the ULTRACOL100 device, comparing it with three commercial products (REJURAN®, JUVELOOK®, and HYRONT®). The ULTRACOL100 device showed comparable performance in improving the appearance of the NLF area among this study subjects.

**Keywords:** nasolabial folds (NLFs); ULTRACOL100; PDO filler; non-surgical facelift

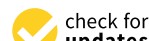



## 1. Introduction

Skin aging contains intrinsic and extrinsic processes. While innately aged skin is inevitable and determined by genomics, extrinsic aging is generated by external modifiable factors such as smoking and sunshine. Clinical signs of cutaneous aging include dryness, irregular pigmentation, loss of elasticity, and wrinkling, prompting patients to seek cosmetic procedures to improve the appearance of their skin [1,2].

A facial wrinkling, or nasolabial fold (NLFs), also known as a smile line, is a structure absent at birth, present at death, and diminished with facial nerve damage. It is one of the earliest signs of aging and is disdained by most people early on in their lives [3].

The nasolabial fold is characterized by structures that support the buccal fat pad and keep it above the fold. NLFs comprise muscle bundles that cross and run parallel to the

fold and fibrous septae that support the fat pad. The muscle around the fold must be separated from its dermis for the fat to descend and soften the folds [4].

Over the years, numerous non-invasive and minimally invasive techniques and strategies have been developed to counteract and manage premature aging, such as cosmetological care, topical products, and invasive procedures (laser rejuvenation, wrinkle correction (anatomical wrinkles), restoration of fat and volume loss, refining contours, etc.) [5–7]. Among the current cosmetic therapies for the treatment of NLFs, dermal filler is the second most popular non-invasive procedure (right after botulinum toxin) used with various filler materials and commercial products on the market [8–10].

The common dermal fillers well-known are hyaluronic acid, with a notable increase in demand up to 30.0% in 2021 [11], followed by other filler materials containing collagen, polymethylmethacrylate (PMMA), calcium hydroxyl apatite, Poly-L-lactic acid (PLLA), polycaprolactone (PCL) [12], and autologous fat [9,13]. Dermal filler demand is continuing to increase, forcing the development of novel filler products and investigating their safety and effectiveness.

Among these, the emergence of tissue restoration devices such as ULTRACOL100 has shown promise in offering a novel approach to facial rejuvenation.

ULTRACOL100 is the first worldwide Polydioxanone (PDO) powder filler approved by the Korea Food and Drug Administration (KFDA). Polydioxanone was first introduced in 1982 as the first biodegradable suture made from the polymer of paradioxanione [14–16]. Polydioxanone (PDO), a poly(ether-ester), is synthesized by the ring-opening polymerization of p-dioxanone. PDO has attracted growing attention in the medical and pharmaceutical fields because of its ability to degrade into low-toxicity monomers inside the body. PDS has a lower modulus compared to polylactic acid (PLA) or polyglycoloic acid (PGA), making it the first degradable polymer used in the production of a monofilament suture [17,18]. PDXOs have been engineered with adjustable physiological and physicochemical characteristics to fulfill stringent requirements for both biodegradability and biocompatibility [14,19]. PDO is also considered a safe material to use in the development of innovative biodegradable medical implants, as it is safer than non-PDO devices [20]. Recently, powdered PDO mixed with sodium carboxymethyl cellulose was developed and considered to be a collagen-inducing material [21].

Additionally, in previous research from our group, a PDO filler-containing device called ULTRACOL100 showed better neocollagenesis, a lower inflammatory response than the hyaluronic acid (HA) filler, and a significant improvement in skin gloss, wrinkles, and density in a small number of five study subjects [22].

Although dermal filler injections are usually considered safe, certain unfavorable occurrences can happen [23]. Clinicians should be skilled in performing infections with the proper method and have in-depth knowledge of any potential inadequate responses [9,24,25].

In this study, we conducted a comprehensive clinical trial to assess the effectiveness and safety profile of ULTRACOL100 in improving skin in the nasolabial fold area in a more significant study subject population (31 individuals). By investigating the effectiveness, potential benefits, and risks of applying the ULTRACOL100 device, our research contributes to the knowledge and guidance for healthcare and individuals seeking effective and safe solutions for nasolabial rejuvenation.

## 2. Materials and Methods

### 2.1. Materials

ULTRACOL100 filler device containing Polydioxanone (PDO) and caboxymethylcellulose Sodium salt (SCMC) from Ultra V Co., Ltd., Seoul, Republic of Korea, was approved by the Korea Food and Drug Administration (KFDA) in 2021. A REJURAN® filler device containing polynucleotide (PN) extracted from salmon milt and hyaluronic acid (HA) was purchased from PharmaResearch Products Co., Ltd., Gyeonggi-do, Republic of Korea. JUVELOOK® is a hybrid filler device containing PDLLA (Poly D, L-lactide) and hyaluronic acid, purchased from BIM Co., Ltd., Chungcheongbuk-do, Republic of Korea. A HYRONT®

filler device containing sodium hyaluronate 25 mg/2.5 mL was purchased from Huvist Pharmaceutical Co., Ltd., Seoul, Republic of Korea.

## 2.2. Subjects and Clinical Investigation

The clinical study was organized and carried out in accordance with GCP (Good Clinical Practice), MFDS (Ministry of Food and Drug Safety) regulations, and Seoul National University Bundang Hospital's standard operating instructions (SOP).

The reliability guarantee was inspected and confirmed by the research director and approved with research numbers HBSE-MGE-22179 (approved on 12 December 2022) and IRB number B-2211-792-003/HBABN01-221219-BR-E0194-01 (approved on 19 December 2022). This study was conducted with 31 volunteers aged 20 to 59 (average age: 33.48 ± 3.53 years) who had wrinkles at the nasolabial fold area (Table 1).

**Table 1.** Characteristics of clinical participants.

| Group | A | B | C |
|---|---|---|---|
| *n* | 10 | 11 | 10 |
| Gender (Male: 1; Female: 2) | 1.40 ± 0.15 | 1.60 ± 0.15 | 1.50 ± 0.16 |
| Age | 38.5 ± 3.00 | 39.27± 1.89 | 33.4 ± 2.29 |
| Commercial device (Left side face area) | REJURAN® | JUVELOOK® | HYRONT® |
| Testing device (Right side face area) | ULTRACOL100 | ULTRACOL100 | ULTRACOL100 |
| Skin type [1] | 3.20 ± 0.46 | 4.10 ± 0.30 | 3.40 ± 0.43 |
| UV exposure [2] | 1.70 ± 0.14 | 1.40 ± 0.15 | 1.40 ± 0.15 |

[1] Skin type: Dry skin: 1; Neutral skin: 2; skin: 3; Complexity skin: 4; Problematic skin. [2] UV exposure: Less than 1 h: 1; 1–3 h: 2; 3 h or more: 3.

## 2.3. Treatments

In this study, the volunteers participated in the research after washing the test area and resting for 20 min in a constant temperature and humidity room (22 ± 2 °C, 50 ± 5%). Afterward, the test product was applied to the nasolabial fold area twice at an interval of four weeks. An anesthetic cream was applied for 30 min before subcutaneously applying the maximum amount of 1 mL or less of the medical material to the testing area using a 25 G sterile needle.

The improvement of the skin characteristics on the nasolabial fold area was evaluated after the application of the ULTRACOL100 device on one facial side and the REJURAN®, JUVELOOK®, or HYRONT® device on another side. The skin restoration was evaluated before the application and four weeks after the first and second applications compared with before the application of the material on each volunteer.

## 2.4. Skin Improvement Evaluation Methods

Measurement of skin texture: Skin texture (roughness) was measured using 3D images of the selected cheek area. The skin texture was measured using a PRIMMOS system program ver. 5.05 (Canfield, OH, USA). Parameter values indicated for skin texture, including roughness (Ra), maximum roughness depth (Rmax), maximum height (Rz), largest positive deviation (Rp), and largest negative deviation (Rv), were analyzed [26]. The average values of skin texture were measured using a 3-dimensional imaging system called PRIMOSCR (Canfield, OH, USA).

Measurement of skin elasticity and internal elasticity: The skin elasticity of the selected cheek area was measured using the Cutometer® MPA580 (C+K, Köln, Germany). 'R1', 'R2', 'R5', and 'R7' parameters (closer to 100%, better elasticity) were analyzed by means of adsorption of skin for three continuous times within 2 s at a constant negative pressure of 450 mbar [27,28].

Measurement of skin moisture level: The amount of moisture in the tested skin area was measured three times using the Corneometer® CM 825 (C+K, Köln, Germany). The average value was analyzed after measurement.

Measurement of moisture content in the skin (skin hydration): The amount of moisture in the skin of the selected cheek area was measured three times using the MoistureMeter D Compact (Delfin, Kuopio, Finland). The average value was analyzed.

Measurement of transepidermal water loss: The average value of the stabilized section was analyzed by measuring the amount of transepidermal water loss in the selected cheek area using Tewameter® HEX (C+K, Köln, Germany).

Measurement of skin tone and gloss: The skin tone (ITA° value: Individual Typology Angle) and gloss (radiance) of the tested area were measured by the optical system of the Spectrophotometer® CM26dG (Minolta, Tokyo, Japan). Each tested skin area was measured twice, and the average value was analyzed.

Measurement of skin transparency: Skin transparency was measured by TMS 1009 (True Systems Co., Ltd., Seongnam, Korea), which employs the principle of polarization goniometry to determine the degree of skin transparency by estimating the quantity of reflected light in the skin by the reflection of irradiation light on the skin. The skin transparency value was measured three times in selected facial areas, and the average value was analyzed.

Measurement of skin pores: The selected facial parts were taken using Antera 3D® CS (Miravex Limited, Dublin, Ireland). The skin pore parameter values were analyzed.

Measurement of skin density: The skin density of the device's applied area was measured by the ultrasonic probe of the DermaLab® Series SkinLab Combo (Cortex Technology, Aalborg, Denmark).

Photo shoot: The photo shoot was taken at three time points, including the time before the application of the medical device, four weeks after the first application, and four weeks after the second application of the medical device. Facial regions were imaged in optical and polarization modes using a VISIA® CR (Canfield, OH, USA).

*2.5. Survey Evaluation of Product's Effectiveness by Study Participants and Skin Safety Assessment*

At each visit and before evaluating the investigators, the volunteers rated their satisfaction with the treatment. The questionnaire on the effectiveness and usability of the product was investigated on a 6-point scale: 1 point: "Not at all", 2 points: "Disagree", 3 points: "I do not think so", 4 points: "I think so", 5 points: "Agree", and 6 points: "Strongly agree". The answers to four to six points were adopted as the positive response rate (%).

For safety evaluation, the researcher observed the subject's test site at each evaluation time point and then confirmed, recorded, and evaluated the condition of the test site through Q&A with the subject. When an adverse reaction occurred due to the product, an adverse reaction report was prepared, and the research director judged the relevance of this to the test product.

*2.6. Statistical Analysis*

All the calculated data in this research were verified for statistical significance using the SPSS Package Program (IBM, Armonk, NY, USA).

The normality of the data were verified using the Shapiro–Wilk test and the Kurtosis and Skewness test. For the evaluation of the before and after application data, a paired t-test was used for parameters, and a Friedman test was used for non-parameter data. Additionally, the Wilcoxon signed rank test and Post-hoc test (Bonferroni correction) were also applied (*, $p < 0.05$).

The homogeneity between groups was analyzed by the paired t-test method, and the statistical significance level was judged to be homogeneous when the prior value between groups is greater than 0.1.

The group comparison of evaluation results is verified by applying the repeated measure ANOVA (RM-ANOVA) if the groups are homogeneous and by applying the analysis covariance (ANVOVA) if they are not homogeneous (†, $p < 0.05$).

The rate of change is calculated as below:

$$\text{Rate of change}(\%) \ = \ \frac{|\text{The value before using} - \text{The value after using}|}{\text{The value before using}} \times 100$$

All figures were drawn using Graph Prism 9 Software (Boston, MA, USA).

## 3. Results

*3.1. The Flow Chart Experiment for Evaluation of Effectiveness in Improvement of the Skin Nasolabial Fold of ULTRACOL100*

In this study, the testing product ULTRACOL100 and commercial products (REJURAN®, JUVELOOK®, and HYRONT®) were applied to each facial side of the nasolabial fold area in a group, as shown in the experimental diagram (Figure 1).

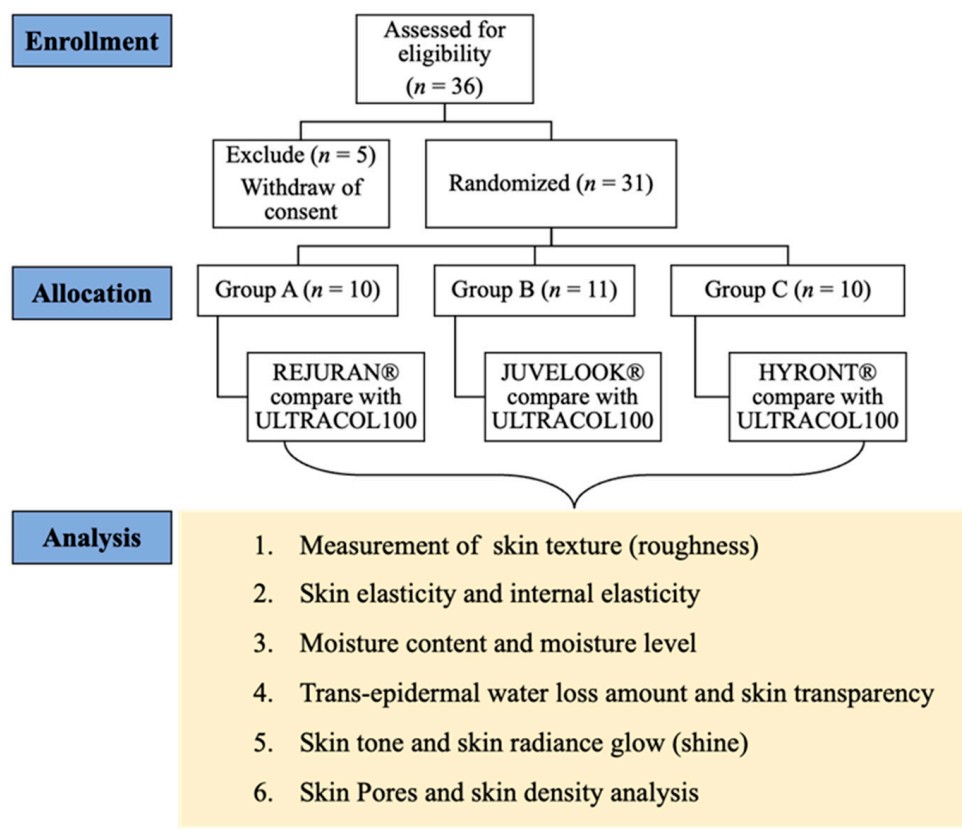

**Figure 1.** Flow diagram of the clinical study.

The volunteers were randomly divided into three groups to apply different commercial products: REJURAN® (Group A; *n* = 10); JUVELOOK® (Group B; *n* = 11); and HYRONT® (Group C; *n* = 10) to compare with ULTRACOL100. The  allocation of study participants into each group was dependent on several factors, such as gender, age, skin type, and UV exposure time. The  baseline characteristics of each clinical participant group are shown in Table 1. The re is no significant difference in any parameter among the participant groups.

The effectiveness of multiple criteria for improving the skin in the nasolabial fold area was evaluated four weeks after twice the application of the testing device, and the results were compared to the conditions prior to using these products and between the different products (Figure 2).

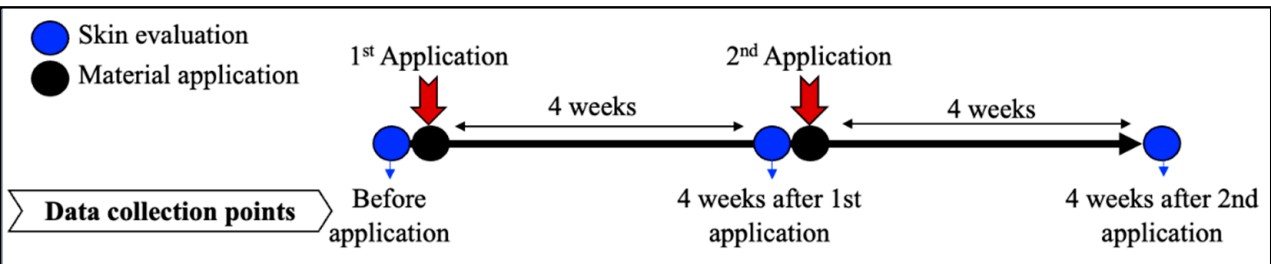

**Figure 2.** Experiment setup and schedule of the visit for evaluation of the device efficiency.

*3.2. The Effectiveness in Improvement of the Skin Characteristics by ULTRACOL100*

3.2.1. Improvement of Skin Roughness

To evaluate the skin texture improvement after the application of each device, the 3-dimensional image system PRISMOSCR® was used to detect skin roughness using a high-resolution sensor. The roughness was indicated in five different parameters, including Average roughness (Ra), Maximum roughness depth (Rmax), Average of the maximum height (Rz), and Largest positive deviation (Rp), Largest negative deviation (Rv). The decrease in these values indicates smoother skin, which was observed in all the ULTRACOL100 testing groups compared with all three control devices after four weeks of the 2nd application (Figure 3). All of the raw data and statistical analysis data were presented in the Supplementary Files, Tables S1 and S2.

In group A, which was tested with commercial REJURAN® and ULTRACOL100 showed better performance in reducing roughness values. After four weeks of each application, there was a notable decrease in Ra (6.72% and 4.23%) and Rz values (6.55% and 3.95%) when compared to the measurement before product usage (* $p = 0.03$ for Ra; and * $p = 0.014$ for Rz). The statistical analysis showed that ULTRACOL100 significantly reduced the Ra († $p = 0.024$), Rz († $p = 0.021$), and Rv († $p = 0.03$) values after four weeks of the 2nd application in comparison with REJURAN® (Figure 3A,B,E,F,I,J).

Furthermore, the performance of the ULTRACOL100 in the JUVELOOK® testing group B also showed a similar result pattern, with a significant decrease observed in all five roughness parameters. After four-weeks of the 1st and 2nd applications of the ULTRACOL100 device, the Ra value decreased by 4.30% and 12.53%, respectively, showing a significant difference (* $p = 0.001$) compared to the prior application of this product. Similar decreases were observed in the Rmax (5.32% and 11.56%), Rz (5.51% and 13.08%), Rp (1.67% and 11.01%), and Rv (9.62% and 13.77%) at four weeks after the 1st and 2nd applications, respectively. In contrast, JUVELOOK® showed no significant improvement in roughness parameters ($p > 0.05$). Detailed values and changes in the roughness parameter values were presented in Group B of Figure 3. Moreover, the application of ULTRACOL100 resulted in a significant decrease in skin roughness four weeks after the 2nd application compared to JUVELOOK®.

In group C, the ULTRACOL100 treatment also decreased the roughness value across all the parameters. The HYRONT® application only showed a reduction in the Rv value, with a slight change of 0.32% and 4.66% after the 1st and 2nd applications, respectively. The statistical analysis indicated that, compared to the application of HYRONT® treatment with ULTRACOL100, all the skin roughness values were higher four weeks after the 2nd application. All of the data were shown in Figure 3, and the pictures of the skin roughness of the represented control group and ULTRACOL100 at the indicated investigation time point were presented in Figure 4.

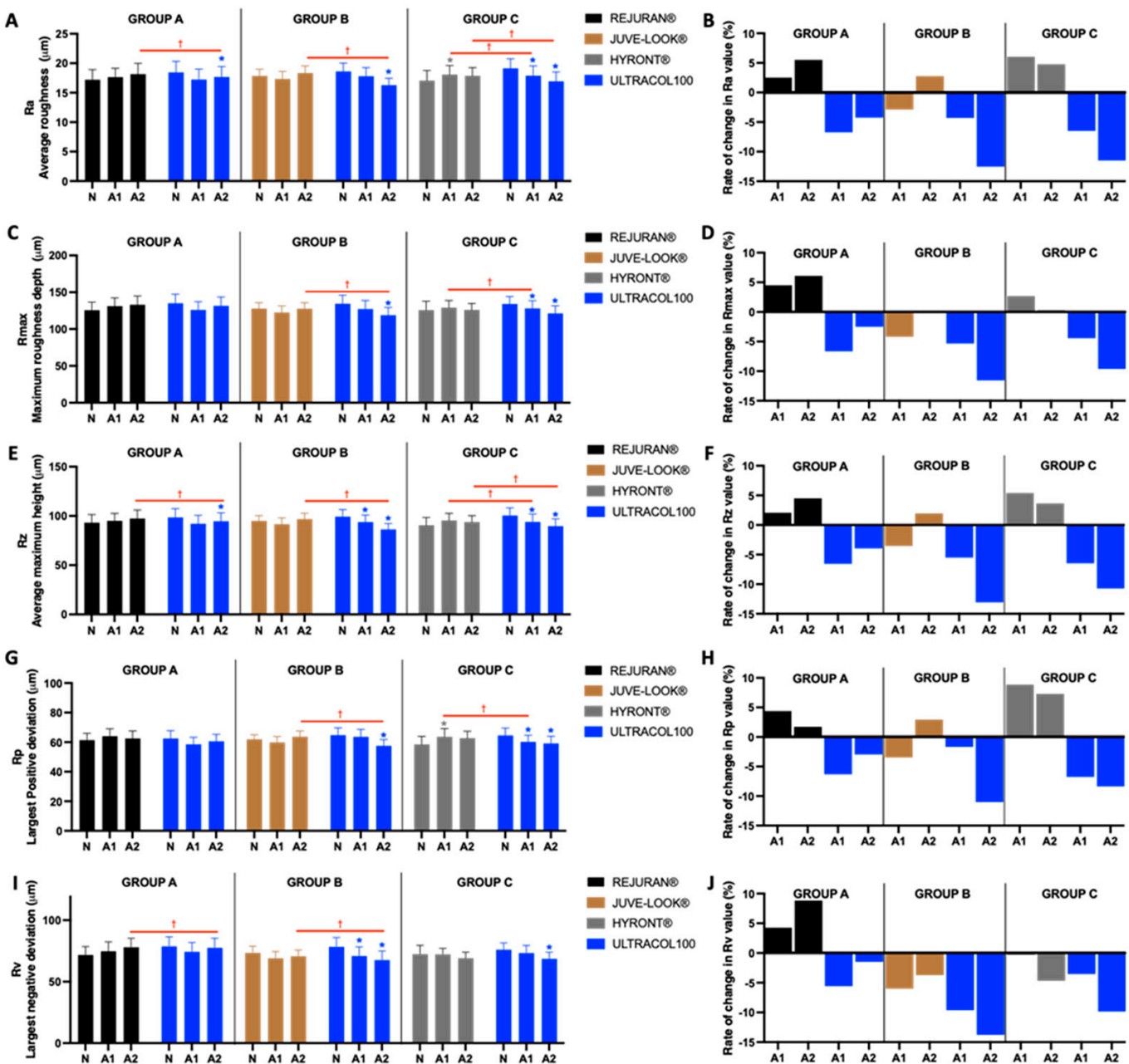

**Figure 3.** The skin texture before and after application of the ULTRACOL100 device in comparison with the commercial products (REJURAN® (GROUP A); JUVELOOK® (GROUP B); and HYRONT® (GROUP C)). The skin texture value of Ra (**A**); Rmax (**C**); Rz (**E**); Rp (**G**); Rv (**I**); and the change in skin texture value of Ra (**B**); Rmax (**D**); Rz (**F**); Rp (**H**); and Rv (**J**) were presented as the average value measured and percentage of the change at four weeks after the 1st and 2nd application of the device in comparison with the skin texture before application. (N: Measuring the skin before the application of the devices; A1: Measuring the skin four weeks after the 1st application of the device; A2: Measuring the skin four weeks after the 2nd application of the device). Data are presented as the mean ± SEM; * $p < 0.05$ with before application of the device; † $p < 0.05$ between each treatment group.

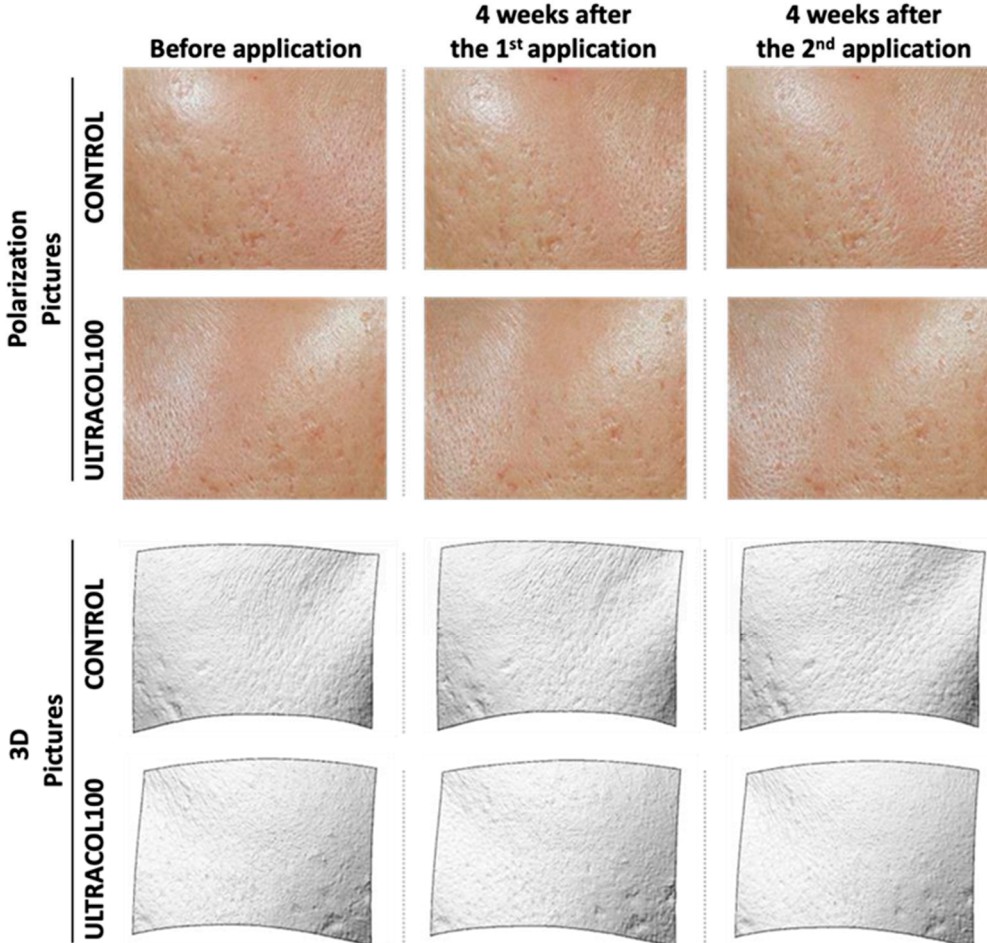

**Figure 4.** Representing polarization pictures and 3D pictures of the skin texture (Roughness) at four weeks after the 1st and 2nd application of the ULTRACOL100 and control device in comparison with before application.

### 3.2.2. Improvement of Skin Elasticity and Skin Firmness

Aging skin is also related to the loss of skin elasticity; therefore, we evaluated skin elasticity after applying the devices using the Cutometer® MPA580, which resulted in releasing four parameters, including R1, R2, R5, and R7. The closer 100% value indicated better elasticity.

The gross elasticity (R2 (%)), which indicates the ability to return against resistance to physical force, increased significantly only in the ULTRACOL100 application among all test groups four weeks after the 2nd application (* $p < 0.05$). The percentage of the increase value was 1.39% and 1.36% in group A, 0.65% and 0.89% for group B, and 1.36% and 1.30% for group C at four weeks after the 1st and 2nd applications, respectively. Furthermore, group B showed a significant increase in gross elasticity with the ULTRACOL100 application († $p = 0.021$) (Figure 5A,B).

The R5 (%) parameter is the ratio of relaxation elasticity to intake elasticity elastic ratio after the first fraction. After the application of ULTRACOL100, the R5 value significantly increased in group B with 6.40% (* $p = 0.026$) at four weeks after the 1st treatment and 6.82% (* $p = 0.023$) at four weeks after the 2nd treatment, and in group C by 5.97% (* $p = 0.019$) after four weeks from the 1st treatment. The R5 value of ULTRACOL100 is significantly higher than that of JUVELOOK at four weeks post- the 1st application († $p = 0.043$) (Figure 5C,D).

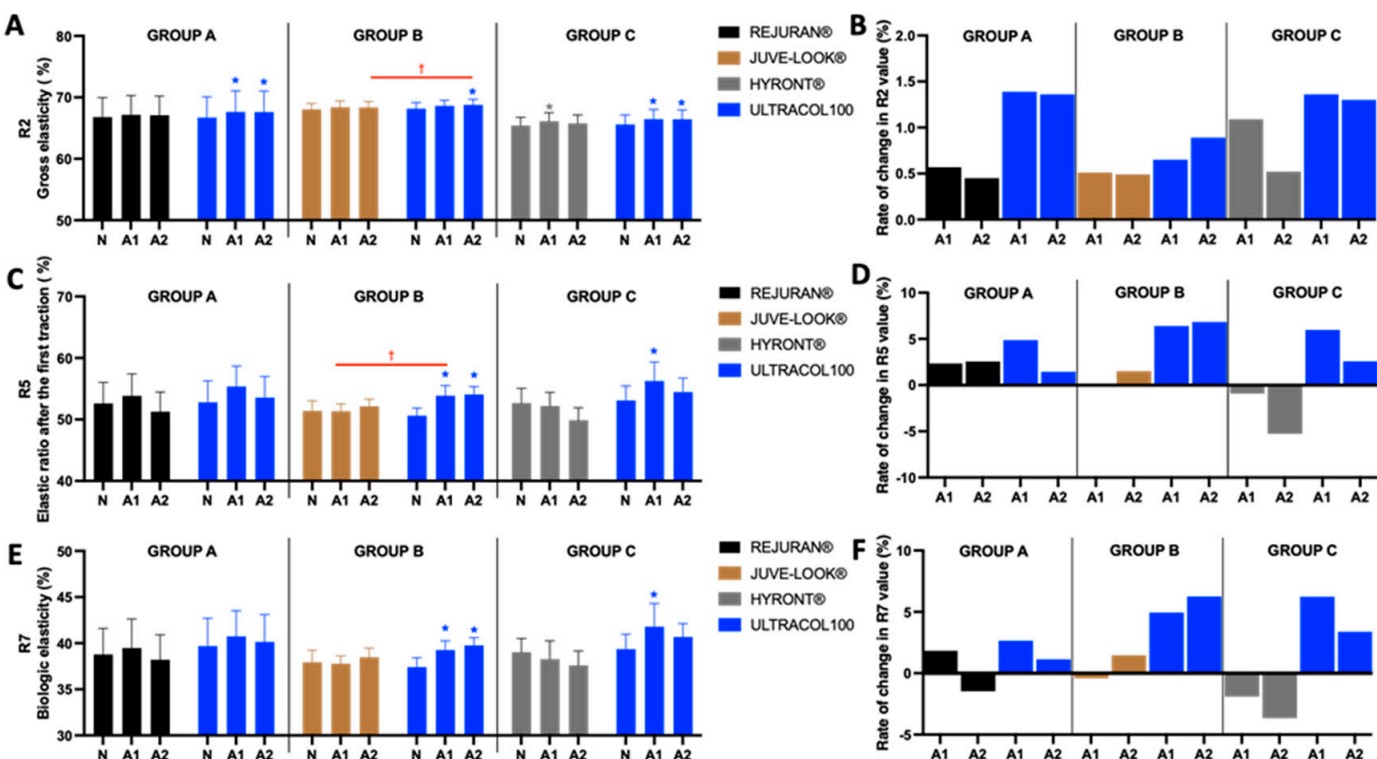

**Figure 5.** The improvement of the skin elasticity after application of the ULTRACOL100 device in comparison with the commercial products (REJURAN® (GROUP A); JUVELOOK® (GROUP B); and HYRONT® (GROUP C)). The skin elasticity values of R2 (**A**), R5 (**C**), and R7 (**E**) and the change in skin elasticity values of R2 (**B**), R5 (**D**), and R7 (**F**) were presented as the average value measured and percentage of the change at four weeks after the 1st and 2nd applications of the device in comparison with the skin elasticity before application. (N: Measuring the skin before the application of the devices; A1: Measuring the skin four weeks after the 1st application of the device; A2: Measuring the skin four weeks after the 2nd application of the device). Data are presented as the mean ± SEM; * $p < 0.05$ with before application of the device; † $p < 0.05$ between each treatment group.

The parameter elasticity R7 (%) represents biological elasticity and the distensibility of the skin, where higher values represent more elastic skin. The graph represents the results over time for each control and ULTRACOL100 testing group. The elasticity measurements increased significantly in ULTRACOL100 treatment groups B and C, with a mean change increase of 4.95 (* $p = 0.006$) and 6.26% (* $p = 0.001$) in group B at four weeks post 1st and 2nd application, respectively, and 6.23% (* $p = 0.039$) in group C after four weeks from the 1st treatment (Figure 5E,F).

The internal elasticity or skin firmness parameter was evaluated by the R1 (mm) value; a minimum amplitude indicates the ability of the skin to return to its original state. The firmer skin showed a significantly lower R1 value (* $p = 0.004$) in the ULTRACOL100-treated group B at four weeks after the 1st application (Figure 6A), with a mean value change of 7.44% compared to before application (Figure 6B). All of the raw data and statistical analysis data were presented in the Supplementary Files Tables S3–S6.

### 3.2.3. Improvement of Skin Moisture Level (Skin Hydration) and Skin Moisture Content

Figure 7 shows the increase in skin moisture content and moisture level (skin hydration) in the ULTRACOL100-applied groups. As shown in Figure 7A, skin hydration significantly increased after each application of the ULTRACOL100 device. In group A, ULTRACOL100 induced a skin hydration increase of 2.09% (* $p = 0.009$) after the 1st application and 6.5% (* $p = 0.021$) after the 2nd application, while REJURAN® showed a slightly non-significant increase of 0.03% ($p = 0.963$%) and 2.19% ($p = 0.194$) after each application (Figure 7A,B).

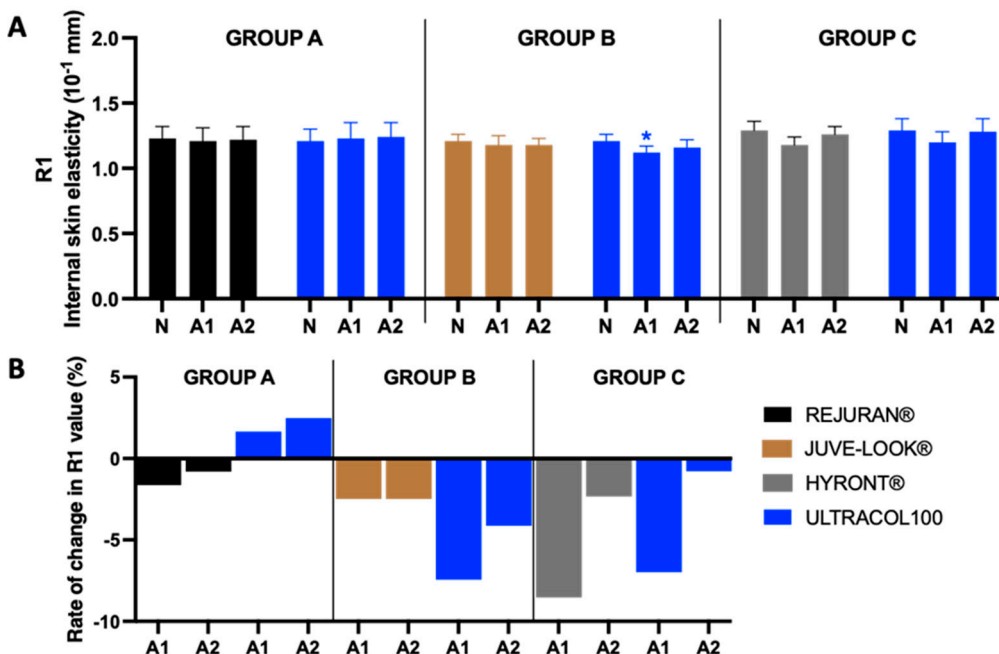

**Figure 6.** The improvement of skin internal elasticity (skin firmness) after application of the ULTRA-COL100 device in comparison with the commercial products (REJURAN® (GROUP A); JUVELOOK® (GROUP B); and HYRONT® (GROUP C)). The skin elasticity value of R1 (**A**) and the change in skin elasticity value of R1 (**B**) were presented in the average value measured and percentage of the change at four weeks after the 1st and 2nd application of the device in comparison with the skin elasticity before application. (N: Measuring the skin before the application of the devices; A1: Measuring the skin four weeks after the 1st application of the device; A2: Measuring the skin four weeks after the 2nd application of the device). Data are presented as the mean ± SEM; * $p < 0.05$ with before application of the device.

A similar pattern was observed in both groups B and C when JUVELOOK® and HYRONT® were applied. ULTRACOL100 significantly induced skin hydration (* $p < 0.05$), while JUVELOOK® showed only a significant increase of 6.41% (* $p = 0.01$%) at four weeks after the 2nd application. Moreover, the statistical analysis demonstrated that ULTRACOL100 significantly increased the skin moisture level in all test groups four weeks after the 2nd application in comparison with all control commercial products († $p < 0.05$) (Figure 7A,B).

The moisture content in group B significantly increased by 1.45% (* $p = 0.009$) after four weeks of the 1st application and by 2.48% (* $p = 0.014$) after the 2nd application, as shown in Figure 7C,D. The statistical analysis indicated that ULTRACOL100 treatment increased the moisture content in all three groups four weeks after the 2nd application († $p < 0.05$) (Figure 7C). All of the raw data and statistical analysis data were presented in the Supplementary Files Tables S7–S10.

### 3.2.4. ULTRACOL100 Treatment Reduces Skin Trans-Epidermal Water Loss and Increases Skin Transparency Value

As shown in Figure 8, a statistically significant decrease in skin trans-epidermal water loss was observed only in the ULTRACOL100-applied subjects, with a decrease of 9.56% (* $p = 0.044$) after four weeks of the 2nd application in group A and 13.38% (* $p = 0.04$) after four weeks of the 1st application in group C (Figure 8A,B). All of the raw data and statistical analysis data were presented in the Supplementary Files Table S11–S12. Additionally, skin transparency was significantly higher 4.72% (* $p = 0.009$) in ULTRACOL100 treatment group A at four weeks after one-time treatment (Figure 8C,D). All of the raw data and statistical analysis data were presented in the Supplementary Files Tables S17 and S18.

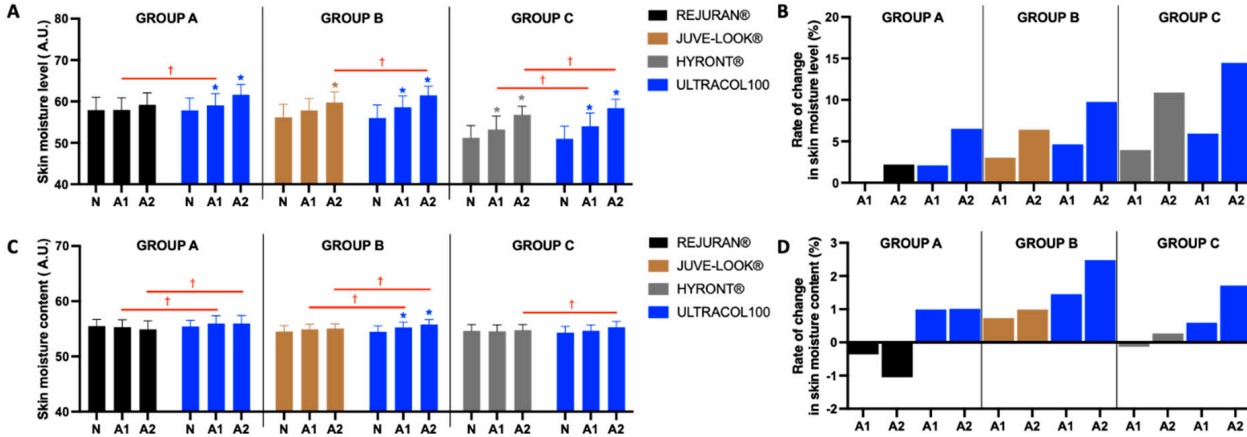

**Figure 7.** The improvement of the skin moisture content and skin moisture level after the application of each device took four weeks. The skin moisture level (**A**), moisture content (**B**), the change in skin moisture level (**C**), and moisture content (**D**) were presented in the average value measured and percentage of the change at four weeks after the 1st and 2nd application of the device in comparison with the skin elasticity before application. (N: Measuring the skin before the application of the devices; A1: Measuring the skin four weeks after the 1st application of the device; A2: Measuring the skin four weeks after the 2nd application of the device). Data are presented as the mean $\pm$ SEM; * $p < 0.05$ with before application of the device; † $p < 0.05$ between each treatment group.

### 3.2.5. ULTRACOL100 Treatment Increases Skin Tone and Skin Radiance

The application of the device led to an increase in skin tone (Figure 9A) and significantly increased at four weeks after 2nd application of two control commercial groups (Group A with REJURANT®: 6.04% (* $p = 0.008$) and Group C with HYRONT®: 6.04% (* $p = 0.008$) and 5.83% (* $p = 0.024$) in the ULTRACOL100 application). Only the ULTRACOL100-treated group showed significant improvement after four weeks of the 1st application, with a rise of 4.72% (* $p = 0.009$) in skin tone value (Individual Typology Angle (ITA°)). All the data for skin tone measurement is shown in Figure 9B,C. All of the raw data and statistical analysis data were presented in the Supplementary Files Tables S13 and S14.

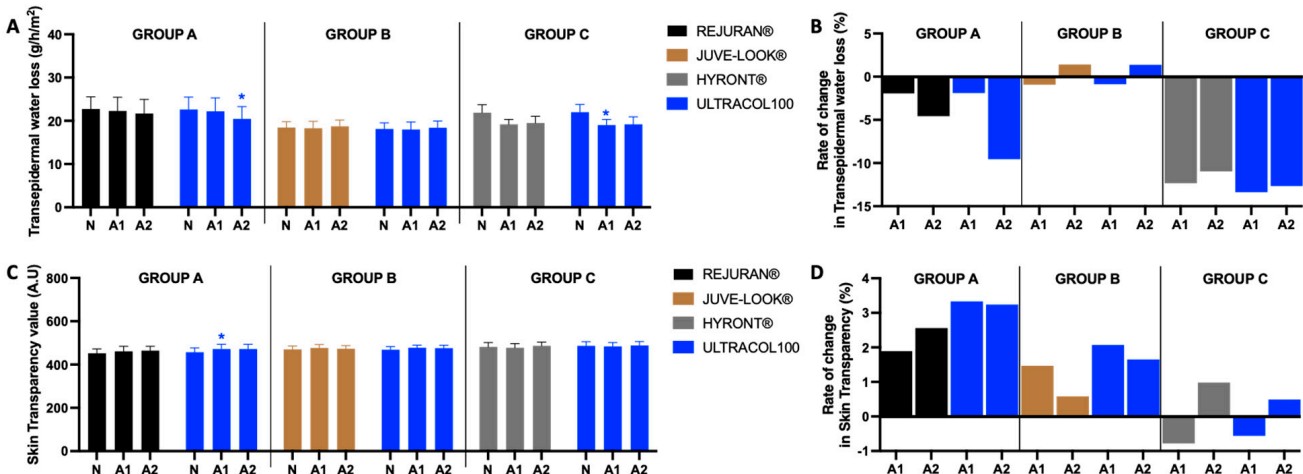

**Figure 8.** The reduction of trans-epidermal water loss and the improvement of skin transparency after the application of each device took four weeks. The trans-epidermal water loss value and the skin transparency value were presented in the average value measured (**A,C**) and percentage of the change (**B,D**) at four weeks after the 1st and 2nd application of the device in comparison with the skin elasticity before application. (N: Measuring the skin before the application of the devices; A1: Measuring the skin four weeks after the 1st application of the device; A2: Measuring the skin four weeks after the 2nd application of the device). Data are presented as the mean $\pm$ SEM; * $p < 0.05$ with before application of the device.

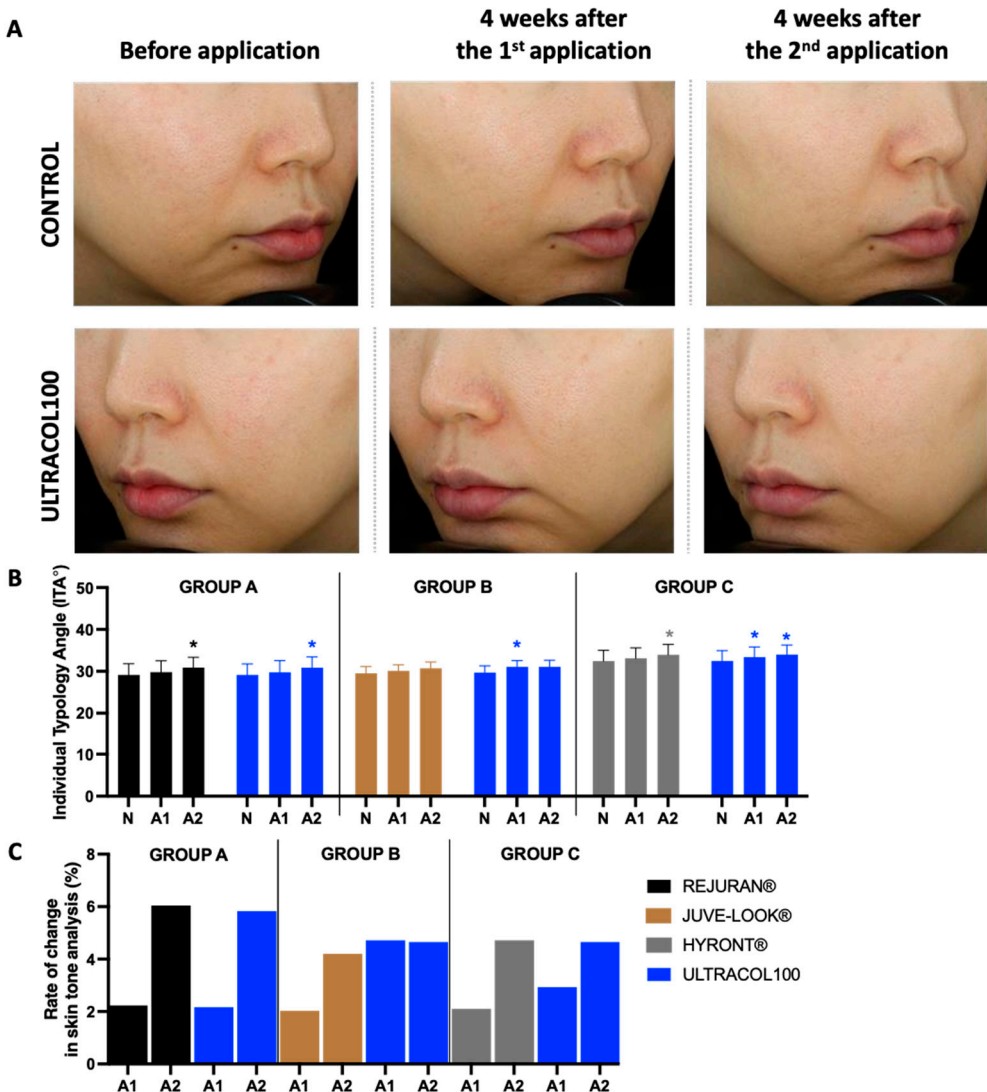

**Figure 9.** The improvement of the skin tone after the application of each device for four weeks. Skin tone pictures at different application time points (**A**), skin tone (Individual Typology Angle (ITA°)) value (**B**), and the change in skin tone value (%) (**C**) were presented. Graphs showed the average value (ITA°) measured and percentage of the change (%) at four weeks after the 1st and 2nd applications of the device in comparison with the skin elasticity before application. (N: Measuring the skin before the application of the devices; A1: Measuring the skin four weeks after the 1st application of the device; A2: Measuring the skin four weeks after the 2nd application of the device). Data are presented as the mean ± SEM; * $p < 0.05$ with before application of the device.

Figure 10 shows the significant improvement of the skin radiance value in all groups of subjects after treatment with ULTRACOL100, while this improvement was only observed in the REJURANT® and JUVELOOK® applied groups. In ULTRACOL100 application subjects, the skin radiance content significantly increased: 12.87% (* $p = 0.001$) in group A, 16.15% (* $p = 0.000$) in group B, 11.83% (* $p = 0.015$) in group C, after four weeks of the 1st application, 18.38% (* $p = 0.003$) in group A, 16.49% (* $p = 0.001$) in group B, and 16.85% (* $p = 0.008$) in group C of the 2nd application, as shown in Figure 10B,C. REJURANT® treatment improves skin radiance only after four weeks of the 2nd application with a rise of 10.26% (* $p = 0.014$) in skin radiance value. JUVELOOK® improved the skin radiance significantly with an evaluation of 9.31% (* $p = 0.01$) and 12,07 (* $p = 0.006$) at four weeks after the 1st and 2nd application, respectively.

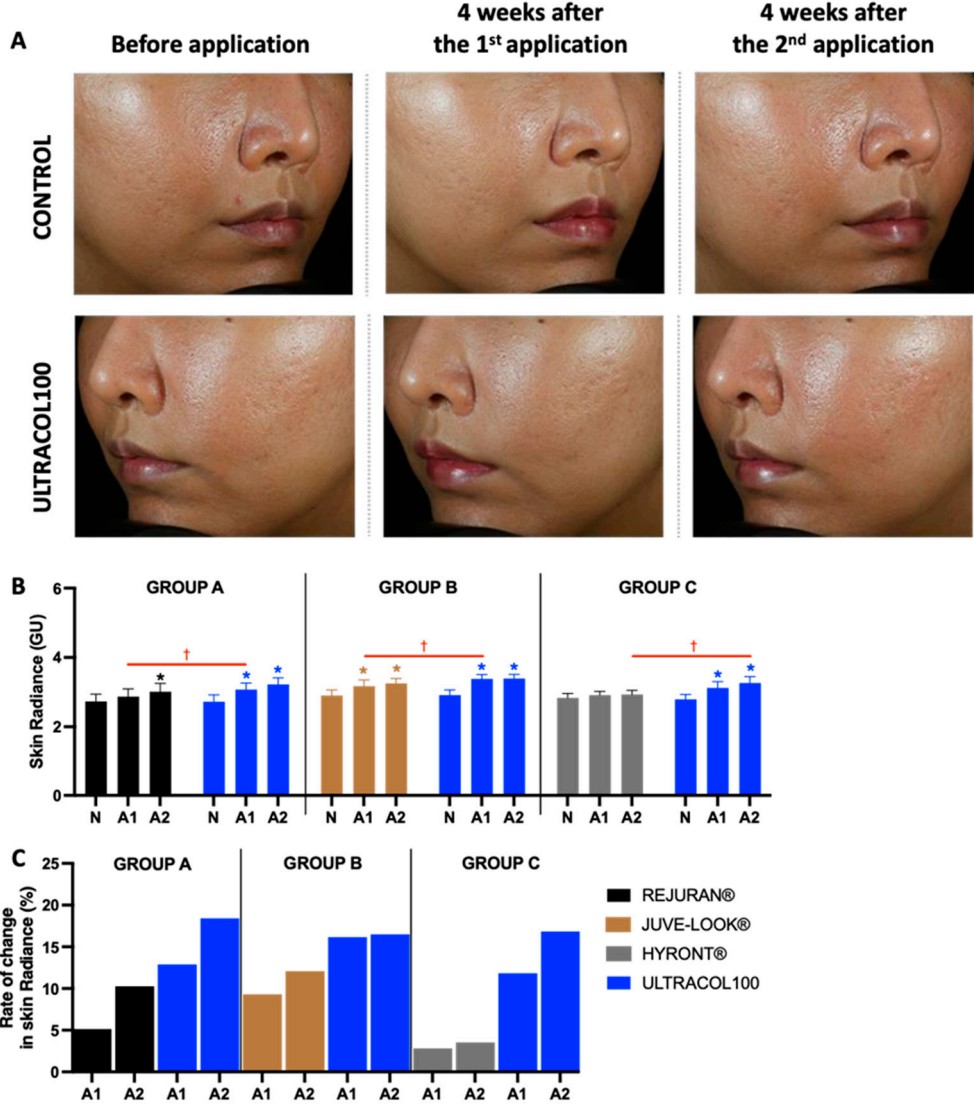

**Figure 10.** The improvement of skin radiance. The skin radiance picture (**A**), the skin radiance value (**B**), and the change in skin radiance value (**C**) were presented. Graphs showed the average value measured and percentage of the change at four weeks after the 1st and 2nd applications of the device in comparison with the skin elasticity before application. (N: Measuring the skin before the application of the devices; A1: Measuring the skin four weeks after the 1st application of the device; A2: Measuring the skin four weeks after the 2nd application of the device). Data are presented as the mean $\pm$ SEM; * $p < 0.05$ with before application of the device; † $p < 0.05$ between each treatment group.

The statistical analysis indicated that ULTRACOL100 treatment increased the skin radiance in groups A and B after four weeks post the 1st application and in group C after four weeks post the 2nd application in comparison with the performance of the control groups († $p < 0.05$) (Figure 10B). All of the raw data and statistical analysis data were presented in the Supplementary Files Tables S15 and S16.

### 3.2.6. Treatment with ULTRACOL100 Reduces the Skin Pores and Improves Skin Density

We evaluated the improvement of the skin in the decrease of the skin pores size using an Antera 3D® CS device, which returned the total pore size area (mm²) value. The smaller the value indicates, the more negligible the skin pores are. As shown in Figure 11, all the device treatments showed a decrease in the pore size; however, the ULTRACOL100-treated subjects significantly decreased the pore size four weeks after both the first and second application. In Group A with REJURANT® treatment control, the

significance was only observed at four weeks after the first application in a decrease of 7.88% (* $p = 0.045$) in REJURANT® application, while ULTRACOL100 decreased pore size significantly from the first application (12.14% (* $p = 0.003$) and 8.67% (* $p = 0.028$) at the second application. Group B showed a significant decrease in only ULTRACOL100 applied compared to before and with the JUVELOOK-treated group four weeks after the second application († $p = 0.017$) (Figure 11B,C). All of the raw data and statistical analysis data were presented in the Supplementary Files Tables S19 and S20.

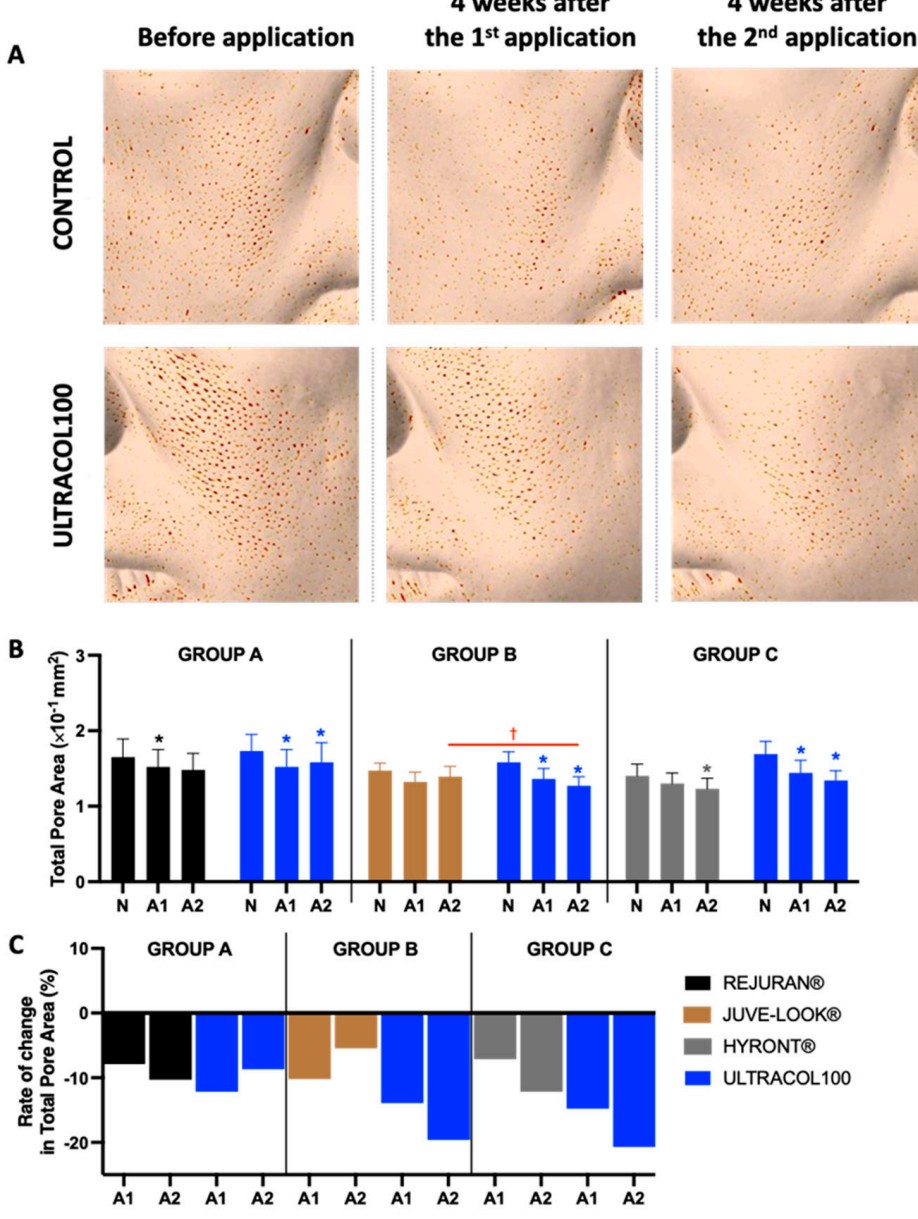

**Figure 11.** The decrease in size of the skin pore after the application of each device. Skin pore pictures (**A**), total skin pore area data (**B**), and the change in total skin pore area values (**C**) were presented in the average value measured and percentage of the change at four weeks after the 1st and 2nd application of the device in comparison with the skin elasticity before application. (N: Measuring the skin before the application of the devices; A1: Measuring the skin four weeks after the 1st application of the device; A2: Measuring the skin four weeks after the 2nd application of the device). Data are presented as the mean ± SEM; * $p < 0.05$ with before application of the device; † $p < 0.05$ between each treatment group.

Following the application of each device to repair nasolabial folds, high-frequency dermal ultrasonography revealed substantial increases in collagen and new collagen density. Figure 12 shows the improvement of skin density (tissue echogenicity) in all device applications. ULTRACOL100 still kept the best performance, with a significant increase in skin density in all the testing subjects from all three groups compared to baseline day 0 (* $p < 0.05$).

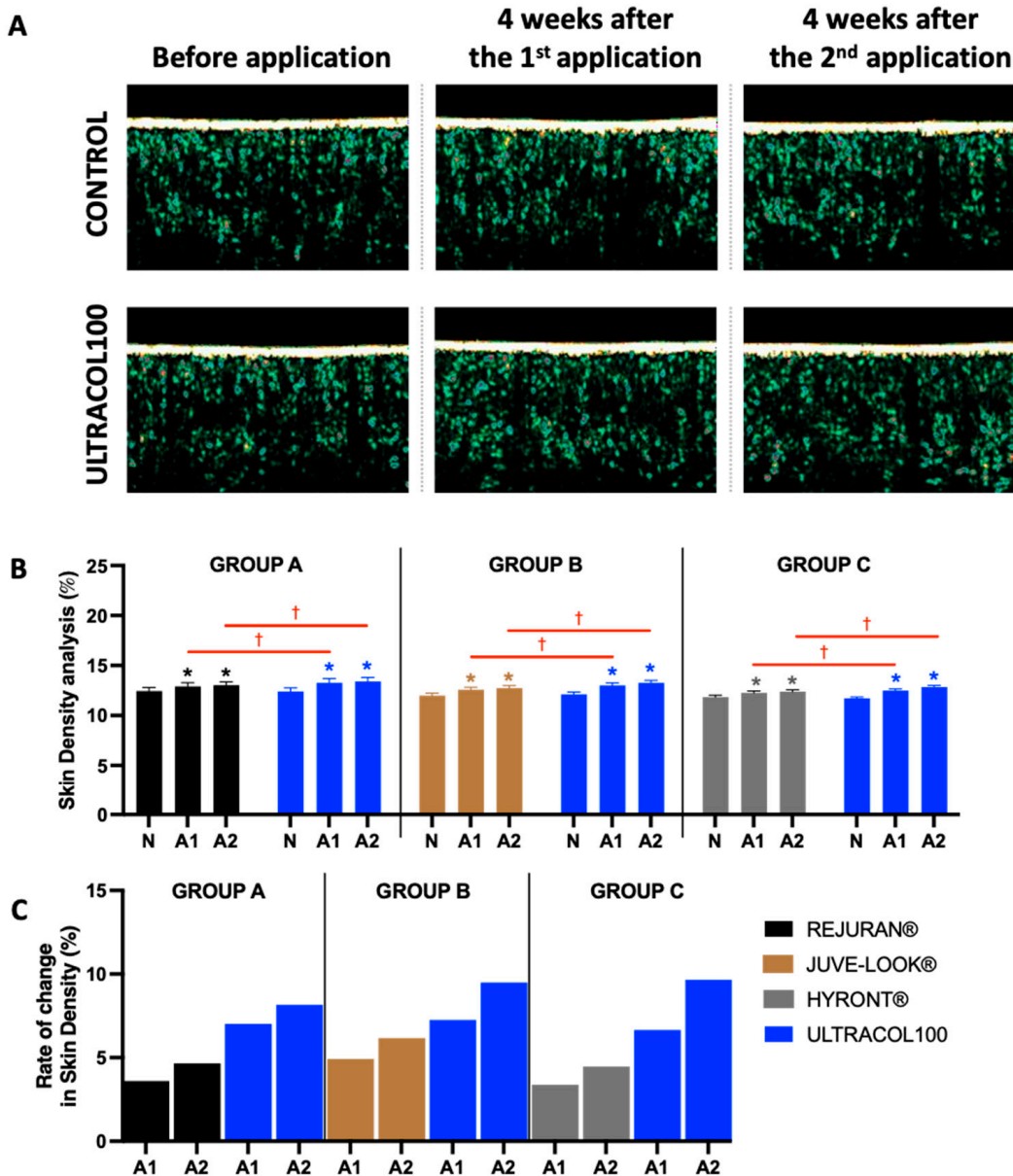

**Figure 12.** The improvement in skin density after the application of each device. Skin density pictures (**A**), skin density analysis value data (**B**), and the change in skin density values (**C**) were presented. Graphs showed the average value measured and percentage of the change at four weeks after the 1st and 2nd applications of the device in comparison with the skin elasticity before application. (N: Measuring the skin before the application of the devices; A1: Measuring the skin four weeks after the 1st application of the device; A2: Measuring the skin four weeks after the 2nd application of the device). Data are presented as the mean ± SEM; * $p < 0.05$ with before application of the device; † $p < 0.05$ between each treatment group.

The statistical analysis indicated that ULTRACOL100 treatment increased the skin density in all groups after four weeks, both the first application and the second application, in comparison with the performance of the control groups († $p < 0.05$) (Figure 12B). All of the raw data and statistical analysis data were presented in the Supplementary Files Tables S21 and S22.

### 3.3. Evaluation Survey by the Research Subjects and Skin Safety Assessment

Four weeks after each application, all research subjects responded to their qualitative evaluation of the product's efficacy following the self-assessment questionnaire, and the results are presented in Table 2. No early study was withdrawn due to treatment-related adverse effects that occurred during the follow-up period. Overall, the questionnaire evaluation of the efficacy and usability of the product received a 61% to 87% positive response.

**Table 2.** Results of the questionnaire evaluation on effectiveness and usability of medical devices at four weeks after the 2nd application of medical devices.

| No. | Questions * | Number (*n*) [1] | Response Rate (%) [2] |
|---|---|---|---|
| 1 | Has the skin texture become smoother after applying the medical device? | 27 | 87.10 |
| 2 | Does the skin volume seem to improve after applying the medical device? | 22 | 70.97 |
| 3 | Does the skin elasticity seem to have improved after applying the medical device? | 24 | 77.42 |
| 4 | Does the deep elasticity of the skin seem to improve after applying the device? | 21 | 67.74 |
| 5 | Does your skin feel moisturized and firm after applying the medical device? | 23 | 74.19 |
| 6 | Does the inner dryness seem to improve after applying the medical device? | 22 | 70.97 |
| 7 | Did you feel that your skin tone improved after applying the medical device? | 23 | 74.19 |
| 8 | Did you feel that your skin radiance (radiance) improved after applying the medical device? | 19 | 61.29 |
| 9 | Did you feel that your skin became transparent after applying the medical device? | 21 | 67.74 |
| 10 | Does it seem to be effective in improving skin pores after applying the medical device? | 20 | 64.52 |
| 11 | Does the skin density seem to have improved after applying the medical device? | 22 | 70.97 |
| 12 | Does it seem to help improve your overall skin after applying the medical device? | 24 | 77.42 |
| 13 | Were you satisfied with the procedure overall? | 27 | 87.10 |
| 14 | Would you recommend this procedure? | 27 | 87.10 |

* Survey scale: 1 point: "Not at all", 2 points: "Disagree", 3 points: "I don't think so", four points: "I think so", 5 points: "Agree", 6 points: "Strong agree". [1] Number (*n*): The number of research subjects who selected 4, 5, or 6 points. [2] Response rate (%): number of research subjects who selected 4, 5, or 6 points/total number of research subjects.

## 4. Discussion

The most notable sign of aging skin is nasolabial folds (NLFs), caused by disruption of the midface muscle contour and decreasing dermal elasticity with an increment of the subcutaneous fatty layer [29].

Aging skin occurs, rendering skin less resilient and elastic and changing pigmentation, pore size, elasticity, oiliness, and thickness [30,31]. The se changes, occurring in conjunction with other age-related modifications and damage, lead to reduced skin barrier function and moisture levels. As a result, these factors contribute to the escalation of problems such as skin sagging, enlarged pores, wrinkles, prominent expression lines, dullness, uneven skin tone, rough texture, excessive pigmentation, dryness, and redness. The refore, we assessed the effectiveness of ULTRACOL100 by evaluating skin quality parameters. Humphrey S. et al. proposed three skin fundamental categories, including visual (tone, radiance, etc.), mechanical (elasticity, firmness, etc.), and topographical (roughness, pores, etc.) [31]. In the current study, we evaluated skin quality via multiple parameters, including skin texture: Roughness, elasticity, and firmness; skin moisture: moisture content; and trans-epidermal water loss; skin appearance: tone, gloss (shine), transparency, pore size, and skin density.

The emergence of wrinkles at nasolabial folds in aging skin can evoke aesthetic distress in many individuals. As a result, many treatment modalities have been developed to enhance their visual appearance. Filler injection therapy is a standard therapeutic and less-invasive cosmetic procedure [32].

ULTRACOL100 is the world's first Polydioxanone (PDO) filler, approved by the KFDA [33]. This product was previously investigated and proved to have a collagen stimulation function in vitro and in vivo and improve skin texture and appearance in a few participants ($n = 5$) [22].

Our research aimed to compare the clinical effects and safety of ULTRACOL100 to commercial control filler products (REJURAN®, JUVELOOK®, and HYRONT®) after eight weeks of treatment in improving the skin in the nasolabial folds area.

The application of the device notably induces restoration of the nasolabial folds with significant improvement of all the criteria of the skin, including skin texture: Roughness, elasticity, firmness (internal elasticity); skin moisture: Moisture content, trans-epidermal water loss; skin appearance: Tone, gloss (shine), transparency, pore size, skin density.

A total of 31 participants were randomly allocated into three groups and treated with one of each control filler product: REJURAN® (group A), JUVELOOK® (Group B), and HYRONT® (Group C). The participant was treated with control filler on one side and ULTRACOL100 on the remaining side of the face at the nasolabial fold's skin area. The improvement of nasolabial rejuvenation of the skin at the smile line area was statistically significantly improved in the ULTRACOL100 treated group (* $p < 0.05$) after four weeks of the second treatment in terms of skin texture, skin elasticity, moisture, density, and skin appearance (tone, glow, radiance, and transparency).

Epidermal fillers have been used for the improvement of deep wrinkles and facial contour by increasing skin volume and rejuvenation [34,35]. A high correlation between skin age, skin roughness parameters, and dermal density was confirmed and proven in several previous studies [36]. Meanwhile, several fillers have shown the ability to induce collagen formation and improve skin thickness [35,37]. In our current study, the effectiveness of ULTRACOL100 on improving skin characteristics was also observed and evaluated on skin texture, also known as roughness, with the decrease of all values of skin roughness (Ra, Rmax, Rz, Rp, Rv) in all UNTRACOL100 treatment groups after four weeks of the first and second application of the device with before treatment (at day 0) (Figure 3).

Aging skin and especially NLFs are intensely relevant to the loss of skin elasticity and subcutaneous fat [29]. In our current study, the significant increase in skin elasticity and improvement of skin firmness were statistically substantial in the ULTRACOL100 application only. The improvement was observed in the rise of the elasticity values (R2, R5, R7) at four weeks after application compared to day 0 (* $p < 0.05$) and to the control groups († $p < 0.05$) (Figure 5). Notably, ULTRACOL100 only induces skin firmness (R1) significantly (* $p < 0.05$) at four weeks post-application (Figure 6).

The difference in the effectiveness aspects of each filler product may come from the divergence of components in each filler product. In detail, ULTRACOL100 contained Polydioxanone (PDO) and carboxymethylcellulose sodium salt (SCMC). Previously, the safety of PDO-containing material in non-surgical facelifts was reported elsewhere as a safe material for facelift thread [38], followed by the first report of PDO as a collagen-stimulating PDO filler material by Kwon T. R. et al. in 2019 [21,32]. Several reports have suggested that PDO also supports facial skin rejuvenation and is beneficial as a material for nonsurgical face lifting. It uses collagen syntheses to induce effects [21,32].

Meanwhile, all the remaining fillers contained hyaluronic acid (HA) and REJURAN, especially supplemented with salmon milt's Polynucleotide (PN), and JUVELOOK, composed of Poly D, L-lactide (PDLLA), and HA; and HYRONT contained hyaluronic acid only. Hyaluronic acid was well known for its filler function, with a low tissue response effect but only a small role in collagen generation [39].

PN contains nutritional effects and the ability to stimulate the secretion of extracellular protein-containing collagens [40]. PDLLA was reported to have the ability to induce fibroblasts to produce collagen and increase cell proliferation, and its unique physiochemical properties also make it controllable and long-lasting [41].

Our data also showed that the skin hydration and moisture level of the ULTRACOL100 treatment were significantly higher after four weeks of each treatment (* $p < 0.05$) and for all control groups († $p < 0.05$) (Figure 7). Although there is no statistically significant difference

in the trans-epidermal water loss, skin transparency, or skin tone of the ULTRACOL100 treatment compared to the control groups († $p > 0.05$) (Figure 8). However, a significant increase in skin tone compared to before application was observed in all ULTRACOL100 treatments together with REJURAN® and HYRONT® at four weeks post-application. In addition, ULTRACOL100 showed a significant increase in skin tone earlier than four weeks after the first application, as data showed in groups B and C of Figure 9 (* $p < 0.05$).

All the product applications improved skin radiance, reduced skin pore size, and increased skin density after application (* $p < 0.05$), and ULTRACOL100 treatment showed statistically significant improvement of each criterion compared to all the control products († $p > 0.05$) (Figures 10–12). All the product applications improved skin radiance, reduced skin pore size, and increased skin density after application (* $p < 0.05$), and ULTRACOL100 treatment showed statistically significant improvement in each criteria compared to all the control products († $p > 0.05$) (Figures 10–12). Overall, the better effectiveness of UL-TRACOL100 treatment may be caused by the characteristics of PDO-based filler, which has previously shown a biostimulator effect. While collagen stimulatory fillers may initially provide less improvement right after treatment compared to other fillers, this improvement will gradually increase. Collagen-boosting fillers like PDO show a little initial improvement in wrinkle reduction; however, over time, this filler will stimulate the production and regrowth of collagen and other connective tissues, which then provide niches and frameworks for fibroblasts or vascular cells to enter. This leads to a higher level of efficacy at a later stage compared to other fillers. Hence, PDO filler is appropriate for those seeking a gradual enhancement [32]. Additionally, it may provide benefits compared to hyaluronic acid- or CaHA-based fillers because of its enhanced stability and long-lasting outcomes [42].

This study also includes evaluations of participant satisfaction and safety as part of a thorough investigation into the effectiveness of ULTRACOL100 for the correction of nasolabial folds (NLFs). Satisfaction and safety were evaluated by the participants and investigators, which returned 61 to 87% positive responses, indicating the promising usability of this product in the market (Table 2).

The relatively brief follow-up period and the small sample size are the main drawbacks of the current study. Despite the considerable improvements in several skin parameters, more research, including a larger subject pool and long-term safety, is still necessary. In the further study, it is advisable to conduct additional toxicological studies, including patch tests, to ensure the safety and compatibility of ULTRACOL100 with a broader population [43].

Therefore, this study would support the ULTRACOL100 intervention's long-term dependability and safety.

## 5. Conclusions

In conclusion, our research has shed light on the potential effectiveness of ULTRACOL100, a Polydioxanone (PDO) filler, in addressing the common signs of aging skin, specifically nasolabial folds (NLFs). Following the ULTRACOL100 treatment, we saw substantial improvements in many skin characteristics, including texture, elasticity, and moisture. Additionally, the safety and satisfaction of the subjects were thoroughly evaluated, resulting in favorable feedback.

Furthermore, our research contributes to the knowledge base and offers helpful advice for anyone looking for safe and effective nasolabial rejuvenation procedures, including healthcare professionals. Although these results show potential for those looking for non-surgical options for nasolabial folds (NLFs), we emphasize the need for more studies with larger sample sizes and longer-term observations to confirm the long-lasting effectiveness and safety of ULTRACOL100.

**Supplementary Materials:** The following supporting information can be downloaded at: https://www.mdpi.com/article/10.3390/cosmetics11010004/s1, Table S1. Statistical analysis of skin texture at before and after application of each device by time point, Table S2. Statistical analysis for comparison between groups of skin texture measurement, Table S3. Statistical analysis of skin elasticity at before and after application of each device by time, Table S4. Statistical analysis for

comparison between groups of skin elasticity measurement, Table S5. Statistical analysis of skin firmness (internal elasticity) at before and after application of each device by time point, Table S6. Statistical analysis for comparison between groups of skin firmness (internal elasticity) measurement, Table S7. Statistical analysis of the skin moisture content at before and after application of each device by time point, Table S8. Statistical analysis for comparison between group of the skin moisture content measurement, Table S9. Statistical analysis of the internal moisture level at before and after application of each device by time point, Table S10. Statistical analysis for comparison between group of the internal moisture level measurement, Table S11. Statistical analysis of the transepidermal water loss at before and after application of each device by time point, Table S12. Statistical analysis for comparison between group of the transepidermal water loss and skin transparency measurement, Table S13. Statistical analysis of skin transparency at before and after application of each device by time point, Table S14. Statistical analysis for comparison between group of the skin transparency measurement, Table S15. Statistical analysis of the skin tone at before and after application of each device by time point, Table S16. Statistical analysis for comparison between group of the skin tone measurement, Table S17. Statistical analysis of the skin radiance at before and after application of each device by time point, Table S18. Statistical analysis for comparison between group of the skin radiance measurement, Table S19. Statistical analysis of the skin pore size at before and after application of each device by time point, Table S20. Statistical analysis for comparison between group of the skin pore size measurement, Table S21. Statistical analysis of the skin density at before and after application of each device by time point, Table S22. Statistical analysis for comparison between group of the skin density measurement

**Author Contributions:** Conceptualization, P.N.C., T.-T.T.T., S.-Y.N. and C.-Y.H.; methodology, T.-T.T.T. and P.N.C.; validation, data curation, T.-T.T.T. and P.N.C.; formal analysis, investigation, L.T.T.L., N.N.-G., and P.T.N.; writing—original draft preparation, P.N.C. and T.-T.T.T.; writing—review and editing, S.-Y.N. and C.-Y.H.; visualization, T.-T.T.T.; supervision, project administration, funding acquisition, S.-Y.N. and C.-Y.H. All authors have read and agreed to the published version of the manuscript.

**Funding:** This study was supported by the Technology Innovation Program (Nanomaterials-based flexible and stretchable sensor system for multimodal monitoring and diagnosis of Sarcopenia, 20015793), funded by the Ministry of Trade, Industry, and Energy (MOTIE, Korea), and was supported by a grant of the Korea Health Technology R&D Project through the Korea Health Industry Development Institute (KHIDI), funded by the Ministry of Health and Welfare, Republic of Korea (grant number: HR22C1363).

**Institutional Review Board Statement:** This study was conducted in accordance with GCP (Good Clinical Practice), MFDS (Ministry of Food and Drug Safety) regulations, and Seoul National University Bundang Hospital's standard operating instructions (SOP). This study was approved with research numbers HBSE-MGE-22179 (approved on 12 December 2022) and IRB number B-2211-792-003/HBABN01-221219-BR-E0194-01 (approved on 19 December 2022).

**Informed Consent Statement:** Informed consent was obtained from all subjects involved in this study.

**Data Availability Statement:** The data presented in this study are available on request from the corresponding author.

**Acknowledgments:** The authors thank Han Jin Kwon, Jung Ryul Ham, Won Ku Lee and Yeon Ju Gu from UltraV Co., Ltd. R&D Center for supplying the materials for the experiment.

**Conflicts of Interest:** The authors declare no conflict of interest.

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
