# Peer review of "An Assessment of the Effectiveness and Safety of ULTRACOL100 as a Device for Restoring Skin in the Nasolabial Fold Region"

_cosmetics, doi:10.3390/cosmetics11010004_

Round 1

Reviewer 1 Report

Comments and Suggestions for Authors

I recommend the manuscript "An assessment of the effectiveness and safety of Ultracol100 as a device for restoring skin in the nasolabial fold region" to be accepted after major revision done. Therefore, I propose:

1. Introduction:

- Page 2, lines 50: it should be indicated what type of wrinkle correction is invasive.

- Repeatedly using hyaluronic acid (HA) instead of the abbreviation.

2. Material and methods

2.1 Materials: why a detailed composition is provided for Hyront® and not for other preparation?

2.3 the description shows that the preparation was administered in an amount of 1 mL in one place, was this really the case?

2.4

-  lines 109: what program are you writing about?

- lines 110: what parameters were measured?

- lines 130-131: There is a lack of description of what and/or how transparency was measured.

3. Results

Point 3.1 - this is a description of the research methodology and the entire material and methods should be included in the section. The authors do not provide the results of their measurements here.

Figure 3 - there are abbreviations which make the content difficult to understand.

4. Discussion

The discussion is mainly a description of the results (lines 451-473), not a comparison of the measurement results with data available in the literature. The discussion is poor in citations and literature references. There is also a lack of self-criticism in the form of a description of the limitations of the presented research results.

5. Conclusion

Conclusions considered bold in relation to a small and undifferentiated study group.

Author Response

Response to Reviewer 1 Comments

1. Summary

We appreciate the distributions and comment for developing our current manuscript. Thank you very much for taking the time to review this manuscript.

Please find the detailed responses below and the corresponding revisions/corrections highlighted/in track changes in the re-submitted files.

2. Questions for General Evaluation

Reviewer’s Evaluation

Response and Revisions

Does the introduction provide sufficient background and include all relevant references?

Can be improved

Thank you for your comments and suggestions. We had corrected and added more information and references, accordingly.

Are all the cited references relevant to the research?

Must be improved

Is the research design appropriate?

Can be improved

Are the methods adequately described?

Can be improved

Are the results clearly presented?

Can be improved

Are the conclusions supported by the results?

Must be improved

3. Point-by-point response to Comments and Suggestions for Authors

1. Introduction:

Comments 1: Page 2, lines 50: it should be indicated what type of wrinkle correction is invasive.

Response 1: Thank you for pointing this out and we agree with this comment. Therefore, we have had included further information of invasive method and made the clear point for this sentence with some additional references.

Revisions 1: Page 2, lines 50: invasive procedures (laser rejuvenation, wrinkle correction (anatomical wrinkles; restoration of fat and volume loss; refining contours), etc.)

Comments 2: - Repeatedly using hyaluronic acid (HA) instead of the abbreviation.

Response 2: We agree with reviewer comment to make it clearer. We have, accordingly, changed all abbreviation of HA into hyaluronic acid.

Revisions 2:

Page 2 Line 96-97: JUVELOOK® is a hybrid filler device containing PDLLA (Poly D, L-lactide) and hyaluronic acid were purchased from BIM Co., Ltd.

Page 17 Line 513-514: HYRONT contained hyaluronic acid only. Hyaluronic acid was well known for its filler function, with a low tissue response effect but only a small role in collagen generation.

2. Material and methods

Comments 3: 2.1 Materials: why a detailed composition is provided for Hyront® and not for other preparation?

Response 3: Thank you for pointing this out, but we could only provide detailed information about Hyront® as the manufacturer provided it publicly on their website. With the remaining products, the detailed amount of each chemical was not provided; therefore, we did not mention it here.

Comments 4: 2.3 the description shows that the preparation was administered in an amount of 1 mL in one place, was this really the case?

Response 4: We appreciate the question and comment from reviewer, therefore we had checked and corrected the sentence to make it clearer and corrected. The maximum amount that could be injected is 1 mL at the nasolabial area of each facial side.

Revisions 4: Page 3 Line 115-117: An anesthetic cream was applied for 30 minutes before subcutaneously applying the maximum amount of 1 mL or less of the medical material to the testing area using a 25G sterile needle.

Comments 5 lines 109: what program are you writing about?

Response 1: Thank you for pointing this out. We agree with this comment, and we had added the information accordingly.

Revisions 1: Page 3, lines 125-128: The skin texture was measured using a PRIMMOS system program (Canfield, USA). Parameter values indicated for skin texture, including roughness (Ra), maximum roughness depth (Rmax), maximum height (Rz), largest positive deviation (Rp), and largest negative deviation (Rv) were analyzed [26].

Comments 6 lines 110: what parameters were measured?

Response 6: Thank you for your correction, we had included the information and rewrote the sentence.

Revisions 6: Page 3, lines 125-128: The skin texture was measured using a PRIMMOS system program (Canfield, USA). Parameter values indicated for skin texture, including roughness (Ra), maximum roughness depth (Rmax), maximum height (Rz), largest positive deviation (Rp), and largest negative deviation (Rv) were analyzed [26].

Comments 7 lines 130-131: There is a lack of description of what and/or how transparency was measured.

Response 7: Thank you for pointing this out. We added the information of transparency measurement accordingly.

Revisions 7: Page 4, lines 149-154: Skin transparency was measured by TMS 1009 (True Systems Co., Ltd, Korea) which employs the principle of polarization goniometry to determine the degree of skin transparency by estimating the quantity of reflected light in the skin by the reflection of irradiation light on the skin. The skin transparency value was measured three times in selected facial areas and the average value was analyzed

3. Results

Comments 8 3.1 - this is a description of the research methodology, and the entire material and methods should be included in the section. The authors do not provide the results of their measurements here.

Response 8: Thank you for the suggestion and strong mention. We provided the information, research methodology, and study regiment; no result is presented in section 3.1. (3.1. The flow chart of experiment for evaluation of effectiveness in improvement of the skin nasolabial fold of ULTRACOL100)

Comments 9 Figure 3 - there are abbreviations which make the content difficult to understand.

Response 9: We appreciated the comment from reviewer for make the manuscript clearer and easier to understand. All of the abbreviations included in the figure 3 was provided in the legend of the figure 3 and the main text at page 6 line 227-229.

4. Discussion

Comments 10 The discussion is mainly a description of the results (lines 451-473), not a comparison of the measurement results with data available in the literature. The discussion is poor in citations and literature references. There is also a lack of self-criticism in the form of a description of the limitations of the presented research results.

Response 10: Thank you for your valuable comment and suggestion developing our current manuscript. We had studied and added the related references and also included the seft-criticism.

Revisions 10:

Page 17, lines 487-495:

Epidermal fillers have been used for improvement of deep wrinkles and facial contour by increase skin volume and rejuvenation [32,33]. High correlation between skin age and skin roughness parameters and dermal density was confirmed and proven in several previous studies [34]. Meanwhile, several fillers have shown the ability to induce collagen formation and improve skin thickness[33,35]. In our current study, the effectiveness of ULTRACOL100 on improving skin characteristics was also observed and evaluated on skin texture, also known as roughness, with the decrease of all values of skin roughness (Ra, Rmax, Rz, Rp, Rv) in all UNTRACOL100 treatment groups after four weeks of the first and second application of the device with before treatment (at day 0) (Figure 3).

Page 17, lines 503-510:

The difference in the effectiveness aspects of each filler product may come from the divergence of components in each filler product. In detail, ULTRACOL100 contained Polydioxanone (PDO) and carboxymethylcellulose sodium salt (SCMC). Previously, the safety of PDO-containing material in non-surgical facelifts was reported elsewhere as a safe material for facelift thread [36], followed by the first report of PDO as a colla-gen-stimulating PDO filler material by Kwon T. R. et al. in 2019 [21,30]. Several reports have suggested that PDO also support facial skin rejuvenation and benefit as a material for nonsurgical face lifting had use collagen systhesis inducing effect [21,30].

Page 18, Line 536-547:

Overall, the better effectiveness of ULTRACOL100 treatment may be caused by the characteristics of PDO-based filler, which has previously shown a biostimulator effect. While collagen stimulatory fillers may initially provide less improvement right after treatment compared to other fillers, this improvement will gradually increase. Colla-gen-boosting fillers like PDO, show a little initial improvement in wrinkle reduction; however, over time, this filler will stimulate the production and regrowth of collagen and other connective tissues, which then provide niches and frameworks for fibroblasts or vascular cells to enter. This leads to a higher level of efficacy in a later stage compared to other fillers. Hence, PDO filler is appropriate for those seeking a gradual enhance-ment[30]. Additionally, it may provide benefits compared to hyaluronic acid- or Ca-HA-based filler because of its enhanced stability and long-lasting outcomes [40].

5. Conclusion

Comments 11 Conclusions considered bold in relation to a small and undifferentiated study group.

Response 11: Thank you for pointing this out, we carefully considered the words and rewrote the conclusions depending on the reviewer’s suggestion.

Revisions 11: Page 18, lines 559-570:

In conclusion, our research has shed light on the potential effectiveness of ULTRACOL100, a Polydioxanone (PDO) filler in addressing the common signs of aging skin, specifically nasolabial folds (NLFs).

Following the ULTRACOL100 treatment, we saw substantial improvements in many skin characteristics, including texture, elasticity, and moisture. Additionally, the safety and satisfaction of the subjects were thoroughly evaluated, resulting in favorable feedback.

Furthermore, our research contributes to the knowledge base and offers helpful advice for anyone looking for safe and effective nasolabial rejuvenation procedures, including healthcare professionals.

Although these results show potential for those looking for non-surgical options for nasolabial folds (NLFs), we emphasize the need for more studies with larger sample sizes and longer-term observations to confirm the long-lasting effectiveness and safety of ULTRACOL100.

Reviewer 2 Report

Comments and Suggestions for Authors

It is a well written manuscript with very good presented results, although, the novelty is rather not too high. The discussion and conclusions are very sufficient and interesting, from the scientific significance point of you. The references are proper, although they are not so many, which is rather explained by a lot of experiments which need a detailed presentation.

Below, some remarks are listed which are necessary for the improvement of manuscript.

It has to be clarified which is the difference between the skin elasticity and skin density, since the increase of elasticity leads to restoration of density. Why do you measure this biophysics parameter ;

3.3:  The safety assessment is almost estimated  by a 24h, 48h, or repeated Patch test. Only the self-assessment is not considered a clear evidence of not causing an adverse event. You have to mention that further toxicological studies such as a Patch test need to be performed in the future.

Comments on the Quality of English Language

It is a well written manuscript with very good presented results, although, the novelty is rather not too high. The discussion and conclusions are very sufficient and interesting, from the scientific significance point of you. The references are proper, although they are not so many, which is rather explained by a lot of experiments which need a detailed presentation.

Below, some remarks are listed which are necessary for the improvement of manuscript.

It has to be clarified which is the difference between the skin elasticity and skin density, since the increase of elasticity leads to restoration of density. Why do you measure this biophysics parameter ;

3.3:  The safety assessment is almost estimated  by a 24h, 48h, or repeated Patch test. Only the self-assessment is not considered a clear evidence of not causing an adverse event. You have to mention that further toxicological studies such as a Patch test need to be performed in the future.

Author Response

1. Summary

We appreciate the distributions and comment for developing our current manuscript. Thank you very much for taking the time to review this manuscript.

Please find the detailed responses below and the corresponding revisions/corrections highlighted/in track changes in the re-submitted files.

2. Questions for General Evaluation

Reviewer’s Evaluation

Response and Revisions

Does the introduction provide sufficient background and include all relevant references?

Yes

Are all the cited references relevant to the research?

Can be improved

Thank you for kindly suggestion. We had rewritten the manuscript and added more references

Is the research design appropriate?

Yes

Are the methods adequately described?

Yes

Are the results clearly presented?

Yes

Are the conclusions supported by the results?

Yes

3. Point-by-point response to Comments and Suggestions for Authors

It is a well written manuscript with very good, presented results, although, the novelty is rather not too high. The discussion and conclusions are very sufficient and interesting, from the scientific significance point of view. The references are proper, although they are not so many, which is rather explained by a lot of experiments which need a detailed presentation.

Below, some remarks are listed which are necessary for the improvement of manuscript.

Response: We thank you for taking your valuable time in reviewer our works. We considered your comments and suggestion carefully and responded to each comment and suggestion as below.

Comments 1: It has to be clarified which is the difference between the skin elasticity and skin density, since the increase of elasticity leads to restoration of density. Why do you measure these biophysics parameter.

Response 1: Thank you for pointing this out. We agree with this comment. Therefore, we have had included further information and the reason why we access these biophysics parameter.

Revisions 1: Page 16 Line 465-476:

Aging skin occurs, rendering skin less resilient and elastic, changing in pig-mentation, pore size, elasticity oiliness, and thickness [30,31]. These changes, occurring in conjunction with other age-related modifications and damage, lead to reduced skin barrier function and moisture levels. As a result, these factors contribute to the escalation of problems such as skin sagging, enlarged pores, wrinkles, prominent expression lines, dullness, uneven skin tone, rough texture, excessive pigmentation, dryness, and redness. Therefore, we assessed the effectiveness of ULTRACOL100 by evaluating skin quality parameters. Humphrey S. et. al. proposed three skin fundamental categories including visual (tone, radiance, etc.), mechanical (elasticity, firmness, etc.), and topographical (roughness, pores, etc.) [31]. In current study, we evaluated skin quality via multiple parameters including skin texture: Roughness, elasticity, firmness; skin moisture: Moisture content, trans-epidermal water loss; skin appearance: Tone, gloss (shine), transparency, pore size, skin density.

Comments 2: 3.3:  The safety assessment is almost estimated  by a 24h, 48h, or repeated Patch test. Only the self-assessment is not considered clear evidence of not causing an adverse event. You have to mention that further toxicological studies such as a Patch test need to be performed in the future.

Response 2: Thank you for a very informative and contributive comment. We had included the information and mentioned it in the discussion.

Revisions 2: Page 18, lines 556-558: In the further study, it is advisable to conduct additional toxicological studies, including patch tests to ensure the safety and compatibility of ULTRACOL100 with a broader population [41].

Reference:

Horita K, Tanoue C, Yasoshima M, Ohtani T, Matsunaga K. Study of the usefulness of patch testing and use test to predict the safety of commercial topical drugs. J Dermatol. 2014 Jun;41(6):505-13. doi: 10.1111/1346-8138.12505. PMID: 24909212.

Reviewer 3 Report

Comments and Suggestions for Authors

The authors present a thorough clinical study on the efficacy of Ultracol 100 to decrease nasolabial folds and improve other skin ageing parameters.

I think the test methodologies used to assess the various skin parameters have been correctly selected and are well described in the paper. Also, data analysis has been performed rigorously, allowing to achieve clear conclusions on the relative performance of the tested products.

Personally, I think that a deeper discussion of the composition and action mechanisms of Ultracol 100 would help to make the paper more scientifically attractive and of broader interest to Cosmetics readers. My main suggestion is to significantly expand the Introduction, by adding more background on Ultracol 100 (what it is and how it is expected to work), and to add more perspective on the different action mechanisms of tested products in the Discussion section.

I also point out that I found the following typos in the text:

Lines 217-218: the sentence is unclear and should be rephrased

Lines 451-452: a sentence terminates abruptedly (either part of the sentence is missing, or it should have been deleted completely).

Line 481: "diversion of components" is not good English

Author Response

Response to Reviewer 3 Comments

1. Summary

We appreciate the comments and suggestions for developing our current manuscript. Thank you very much for taking the time to review this manuscript. We found your comments and suggestions to be supportive and contributive to completing our current study. Therefore, we have carefully revised points to points as per your suggestion.

Please find the detailed responses below and the corresponding revisions/corrections highlighted/in track changes in the re-submitted files.

2. Questions for General Evaluation

Reviewer’s Evaluation

Response and Revisions

Does the introduction provide sufficient background and include all relevant references?

Yes

Are all the cited references relevant to the research?

Can be improved

We appreciate the comment from reviewer for developing our current manuscript. We had revised and added more relevant references.

Is the research design appropriate?

Yes

Are the methods adequately described?

Yes

Are the results clearly presented?

Yes

Are the conclusions supported by the results?

Yes

3. Point-by-point response to Comments and Suggestions for Authors

The authors present a thorough clinical study on the efficacy of Ultracol 100 to decrease nasolabial folds and improve other skin ageing parameters.

I think the test methodologies used to assess the various skin parameters have been correctly selected and are well described in the paper. Also, data analysis has been performed rigorously, allowing to achieve clear conclusions on the relative performance of the tested products.

Comments 1: Personally, I think that a deeper discussion of the composition and action mechanisms of Ultracol 100 would help to make the paper more scientifically attractive and of broader interest to Cosmetics readers. My main suggestion is to significantly expand the Introduction, by adding more background on Ultracol 100 (what it is and how it is expected to work), and to add more perspective on the different action mechanisms of tested products in the Discussion section.

Response 1: We appreciate the suggestion and review point from reviewer, we had expanded the introduction with background on ULTRACOL100 and discuss more about the different action mechanisms of tested products as the revision added below.

Revisions 1:

Adding more background on ULTRACOL100 in introduction (what it is and how it is expected to work):

Introduction

Page 2, lines 63-80:

ULTRACOL100 is the worldwide first Polydioxanone (PDO) powder filler approved by the Korea Food and Drug Administration (KFDA). Polydioxanone was first introduced in 1982, as the first biodegradable suture made from polymer of paradioxanione [14–16]. Polydioxanone (PDO), a poly(ether-ester), is synthesized by the ring-opening polymerization of p-dioxanone. PDO has attracted growing attention in the medical and pharmaceutical fields because of its ability to degrade into low-toxicity monomers inside the body. PDS has a lower modulus compared to polylactic acid (PLA) or polyglycoloic acid (PGA), making it the first degradable polymer used in the production of a monofilament suture [17,18]. PDXOs have been engineered with adjustable physiological and physi-cochemical characteristics to fulfill stringent requirements for both biodegradability and biocompatibility [14,19]. PDO is also considered a safe material to use in the development of innovative biodegradable medical implants as it is safer than non-PDO devices [20]. Recently, powdered PDO mixed with sodium carboxymethyl cellulose was developed and considered to be a collagen-inducing material [21].

Additionally, in previous research from our group, PDO filler containing device as ULTRACOL100 showed better neocollagenesis, a lower inflammatory response than the hyaluronic acid (HA) filler, and a significant improvement of skin gloss, wrinkles and density in a small number of five study subjects [22].

Discussion:

Perspective on the composition and different action mechanisms of ULTRACOL100:

Page 17, lines 501-509:

Epidermal fillers have been used for improvement of deep wrinkles and facial contour by increase skin volume and rejuvenation [34,35]. High correlation between skin age and skin roughness parameters and dermal density was confirmed and proven in several previous studies [36]. Meanwhile, several fillers have shown the ability to induce collagen formation and improve skin thickness[35,37]. In our current study, the effec-tiveness of ULTRACOL100 on improving skin characteristics was also observed and evaluated on skin texture, also known as roughness, with the decrease of all values of skin roughness (Ra, Rmax, Rz, Rp, Rv) in all UNTRACOL100 treatment groups after four weeks of the first and second application of the device with before treatment (at day 0) (Figure 3). 

Page 17, lines 517-524:

The difference in the effectiveness aspects of each filler product may come from the divergence of components in each filler product. In detail, ULTRACOL100 contained Polydioxanone (PDO) and carboxymethylcellulose sodium salt (SCMC). Previously, the safety of PDO-containing material in non-surgical facelifts was reported elsewhere as a safe material for facelift thread [38], followed by the first report of PDO as a colla-gen-stimulating PDO filler material by Kwon T. R. et al. in 2019 [21,32]. Several reports have suggested that PDO also support facial skin rejuvenation and benefit as a material for nonsurgical face lifting had use collagen syntheses inducing effect [21][32]

Page 18 Line 550-560:

Overall, the better effectiveness of ULTRACOL100 treatment may be caused by the characteristics of PDO-based filler, which has previously shown a biostimulator effect. While collagen stimulatory fillers may initially provide less improvement right after treatment compared to other fillers, this improvement will gradually increase. Colla-gen-boosting fillers like PDO, show a little initial improvement in wrinkle reduction; however, over time, this filler will stimulate the production and regrowth of collagen and other connective tissues, which then provide niches and frameworks for fibroblasts or vascular cells to enter. This leads to a higher level of efficacy in a later stage compared to other fillers. Hence, PDO filler is appropriate for those seeking a gradual enhancement [32]. Additionally, it may provide benefits compared to hyaluronic acid- or Ca-HA-based filler because of its enhanced stability and long-lasting outcomes [42]

Comments 2: Lines 217-218: the sentence is unclear and should be rephrased

Response 2: Thank you for pointing this out. We agree with this comment. Therefore, we had rewritten the sentence accordingly.

Revisions 2: Page 6, lines 245-246: In group A, which was tested with commercial REJURAN®, ULTRACOL100 showed better performance in reducing roughness values. After four weeks of each application, there was a notable decrease in Ra (6.72% and 4.23%) and Rz values (6.55% and 3.95%) when compared to the measurement before product usage

Comments 3: Lines 451-452: a sentence terminates abruptedly (either part of the sentence is missing, or it should have been deleted completely).

Response 3: We appreciate reviewer’s correction. We removed the redundant part and rewrote the sentence to make it more clarity.

Revisions 3: Page 16, lines 481-483: Our research aimed to compare the clinical effects and safety of the ULTRACOL100 to commercial control filler products (Rejuran®, Juvelook®, and Hyront®) after eight weeks of treatment in improving the skin at nasolabial folds area.

Comments 4: "diversion of components" is not good English

Response 4: Thank you for kindly pointing this out. We corrected the word accordingly.

Revisions 4: Page 17, lines 515-516: The difference in the effectiveness aspects of each filler product may come from the divergence of components in each filler product.
